# Systemic Administration of PTH Supports Vascularization in Segmental Bone Defects Filled with Ceramic-Based Bone Graft Substitute

**DOI:** 10.3390/cells10082058

**Published:** 2021-08-11

**Authors:** Holger Freischmidt, Jonas Armbruster, Emma Bonner, Thorsten Guehring, Dennis Nurjadi, Maren Bechberger, Robert Sonntag, Gerhard Schmidmaier, Paul Alfred Grützner, Lars Helbig

**Affiliations:** 1Department of Trauma and Orthopedic Surgery, BG Trauma Center Ludwigshafen at Heidelberg University Hospital, 67071 Ludwigshafen am Rhein, Germany; holger.freischmidt@bgu-ludwigshafen.de (H.F.); jonas.armbruster@bgu-ludwigshafen.de (J.A.); emma.bonner@live.at (E.B.); Paul.Gruetzner@bgu-ludwigshafen.de (P.A.G.); 2Trauma Centre, Hospital Paulinenhilfe Stuttgart at Tübingen University Hospital, Rosenbergstr. 38, 70176 Stuttgart, Germany; thorsten.guehring@diak-stuttgart.de; 3Department of Infectious Diseases Medical Microbiology and Hygiene, Heidelberg University Hospital, Im Neuenheimer Feld 324, 69120 Heidelberg, Germany; dennis.nurjadi@med.uni-heidelberg.de; 4Pharmacy Department, Heidelberg University Hospital, Im Neuenheimer Feld 672, 69120 Heidelberg, Germany; maren.bechberger@med.uni-heidelberg.de; 5Laboratory of Biomechanics and Implant Research, Clinic for Orthopedics and Trauma Surgery, Heidelberg University Hospital, Schlierbacher Landstrasse 200a, 69118 Heidelberg, Germany; robert.sonntag@med.uni-heidelberg.de; 6Clinic for Orthopedics and Trauma Surgery, Center for Orthopedics, Heidelberg University Hospital, Schlierbacher Landstrasse 200a, 69118 Heidelberg, Germany; Gerhard.Schmidmaier@med.uni-heidelberg.de

**Keywords:** animal model, non-union, bone defect, parathyroid hormone, ceramic-based bone graft substitute

## Abstract

Non-unions continue to present a challenge to trauma surgeons, as current treatment options are limited, duration of treatment is long, and the outcome often unsatisfactory. Additionally, standard treatment with autologous bone grafts is associated with comorbidity at the donor site. Therefore, alternatives to autologous bone grafts and further therapeutic strategies to improve on the outcome and reduce cost for care providers are desirable. In this study in Sprague–Dawley rats we employed a recently established sequential defect model, which provides a platform to test new potential therapeutic strategies on non-unions while gaining mechanistic insight into their actions. The effects of a combinatorial treatment of a bone graft substitute (HACaS+G) implantation and systemic PTH administration was assessed by µ-CT, histological analysis, and bio-mechanical testing and compared to monotreatment and controls. Although neither PTH alone nor the combination of a bone graft substitute and PTH led to the formation of a stable union, our data demonstrate a clear osteoinductive and osteoconductive effect of the bone graft substitute. Additionally, PTH administration was shown to induce vascularization, both as a single adjuvant treatment and in combination with the bone graft substitute. Thus, systemic PTH administration is a potential synergistic co-treatment to bone graft substitutes.

## 1. Introduction

Non-unions form a severe complication in the fracture healing process [1]. Even though fractures typically tend to heal well and regain full mechanical stability due to the complex intrinsic repair mechanisms [2], delayed or impaired fracture healing, often associated with large bone defects and site infection, occurs in about 5–10% of cases [3,4].

The therapy of delayed or non-union healing poses major challenges to the orthopedic and trauma surgeon and carries an uncertain outcome for the patient [5]. Standard treatment relies on repeated surgical debridement, adequate fixation, reconstruction of the defect, and antibiotic regimen [5]. The gold standard for reconstruction is autologous bone grafting, but these grafts are of limited availability, and donor site morbidity remains an issue [3]. Furthermore, these procedures are often associated with long hospitalization periods leading to adverse professional, social, and financial consequences for the patients, as well as a high burden on the health care system [6]. 

Considering the poor outcome of the current therapeutic methods and the bleak outlook for affected patients, it becomes clear that new approaches are urgently needed to improve orthopedic care. One such new approach is the promising possibility of ceramic-based bone graft substitutes. They assist bone healing by providing an osteoconductive matrix. Additionally, they could easily be combined with a pharmacological treatment to increase osteoconductive stimulus or influence the cell population.

So far, many different materials and material combinations have been tested as bone graft substitutes [7]. Cerament G (Bonesupport, Lund, Sweden) is a resorbable bone graft substitute combining calcium sulphate, hydroxyapatite, and gentamicin [8]. The combination of both seems to provide optimal conditions for bone healing [7,8]. Furthermore, the gentamicin component provides local antibiotic administration, leading to high concentrations at the site and fewer systemic complications than systemic administration [9,10], perhaps reducing the need for additional aggressive systemic antibiotic therapy.

This specific bone graft substitute has shown promising outcomes in animal studies [11,12] as well as in humans [8,10,13,14].

Large clinical trials certainly remain the gold standard testing format. However, detailed analysis of osteogenesis and biomechanical properties among different treatment approaches is not possible in clinical settings. Furthermore, heterogeneity of trauma patients complicates randomized comparative studies. Here, animal models offer a more reliable approach, as experimental parameters can be controlled better. Therefore, further studies on animal models are warranted in order to gather reliable data on the underlying mechanisms of fracture healing respective to the treatment method. The sequential animal model for non-unions with segmental bone defects as described previously facilitates the process of evaluating and comparing bone graft substitutes [13]. Furthermore, it enables reproducible investigation of additional treatment options.

Positive qualities of bone graft substitutes can be supported by pharmacotherapy. A favorable option for additional pharmacological treatment is the systemic application of parathyroid hormone (PTH). PTH is known to have positive effects in osteoporotic patients, as it increases bone mineral density and decreases fracture risk, and is already approved for clinical use in this context [15,16].

Several experimental studies on animal models show an enhancing effect of PTH on fracture healing [17,18,19,20,21] and several case reports have also demonstrated possible positive effects of PTH on non-unions in humans [22,23].

To our knowledge, no study to date has tested a combination treatment of PTH and a ceramic-based bone graft substitute in non-unions.

The aim of the present study is to investigate the effects of a combination treatment of non-unions consisting of systemic parathyroid hormone application and defect filling by a ceramic-based bone graft substitute emitting gentamicin. We employ the abovementioned, already established non-union model on rats. This model includes a unique two-stage approach, which reflects more accurately the clinical reality of non-unions. We hypothesize that PTH combined with bone substitute has a positive effect on bone healing and vascularization in the treatment of non-union. We thereby hope to improve data on alternative treatment methods for complicated fracture healing.

## 2. Materials and Methods

### 2.1. Calcium Sulphate/Hydroxyapatite Bone Graft Substitute with Gentamicin (HACaS+G)

Cerament G (Bonesupport Holding, Lund, Sweden) is a synthetic bone graft substitute consisting of a powder component (40% hydroxyapatite and 60% calcium sulfate) and a liquid component (saline and gentamicin), with the final concentration of gentamicin being 17.5 mg/mL paste. The paste was prepared directly before administration, following the instructions of the manufacturer.

### 2.2. Parathormone

A 250 µg/mL parathormone solution obtained from a Forteo 20 µg/80 µL Teriparatide injector (Eli Lilly Nederland B.V., Utrecht, The Netherlands) was further diluted with 0.9% saline solution to a final concentration of 0.6 µg/mL. This solution was prepared daily immediately before subcutaneous administration.

### 2.3. Implants

Three different kinds of implants, stainless steel Kirschner wires (K-wires) with a diameter of 1.2–1.6 mm (Synthes GmbH, Umkirch, Germany) were used for the first osteosynthesis and an angle-stable plate-osteosynthesis system (RISystem AG, Davos, Switzerland) for the second step procedure. The plate-osteosynthesis system consisted of an angle-stable polyacetyl plate (length, 25 mm; width, 4 mm; height, 4 mm) and 6 cortex screws.

The following groups were examined:Control group (*n* = 15): K-wire osteosynthesis, 5 weeks later re-osteosynthesis with an angle-stable plate, followed by 6 weeks of once-daily administration of parathormone.Intervention group (*n* = 19): K-wire osteosynthesis, 5 weeks later re-osteosynthesis with an angle-stable plate and introduction of HACaS+G into the defect, followed by 6 weeks of once-daily administration of parathormone.

To evaluate the effect of PTH separately, two other groups were included in the analysis. One group was solely treated with HACaS+G, and another group had no treatment. The data for these groups were generated identically to our procedure.

### 2.4. Animals, Operative Procedure, and Osteotomy Model

All experiments were approved by the Animal Experimentation Ethics Committee of Karlsruhe (35-9185.81/G-155/17). Thirty-four female, 3-month-old Sprague–Dawley rats (Charles River, Sulzfeld, Germany) were studied. The animals were maintained on a 12 h light/12 h dark cycle at approximately 22 °C and fed ad libidum. According to size and wellbeing of the animals, between 1 and 5 animals were kept in the same cage.

To test the combination treatment of parathormone and a calcium sulphate/hydroxyapatite composition eluding gentamicin as a bone graft substitute, an already established two-stage animal non-union model was used [13].

Briefly, a full thickness defect on the femora was created by osteotomy during a first intervention, stabilized with a K-wire, and after five weeks, in a second intervention, the developed non-union was treated with an angle-stable plate osteosynthesis.

Surgery was performed under general anesthesia by weight-adapted subcutaneous injection of medetomidine (Dorbene Vet 1 mg/mL; Pfizer Deutschland GmbH, Berlin, Germany), midazolam (Midazolam HEXAL 5 mg/1 mL; Hexal AG, Holzkirchen, Germany), and fentanyl (Fentadon Dechra 50 µg/mL; Dechra Veterinary Products Deutschland GmbH, Aulendorf, Germany).

In the first intervention, a 5 mm mid-diaphyseal full-thickness defect, created by an osteotomy, was performed on the left femora using a diamond disk (Dremel, Racine, WI, USA). The defect was then stabilized with a 1.2–1.6 mm stainless steel K-wire (Synthes GmbH, Umkirch, Germany), resulting in an unstable rotational osteosynthesis. The thickness of the K-wire was adapted according to the diameter of the medullary cavity.

During the second intervention, the K-wire was removed, and the bone defect was radically debrided. Microbiological swabs were taken from the non-union region. Afterwards, an angle-stable polyacetyl plate (RISystem AG, Davos, Switzerland) was placed on the anterolateral surface of the femur and fixated with 6 angle-stable cortex screws to ensure fixture of the approximately 5 mm defect.

In the intervention group, after placement of the six angle-stable screws, the soft tissue was dried, and HACaS+G (Bonesupport Holding, Lund, Sweden) was prepared according to the instructions of the manufacturer. A total of 1.85 g hydroxyapatite/calcium sulphate powder and 0.8 mL gentamicin solution were combined and mixed thoroughly. The composition was left to thicken for about 1 to 2 min. When the desired consistency was reached, about 0.7 mL paste were administered in the defect, bridging the 5 mm gap between the two bone ends fixated before. Thereafter, in both groups, the soft tissue was irrigated, and the wound was closed as described before [13].

### 2.5. Follow-Up

After the surgery, body weight was checked daily for as long as it took the animals to regain their preoperative bodyweight. The wellbeing of the animals (swelling, reddening, impairment of wound healing, loss of passive motion in the left hind leg) was checked daily by a scoresheet.

All animals received buprenorphine (0.3 mg/mL bw; Buprenovet^®;^ Bayer AG, Leverkusen, Germany) as analgesic medication perioperatively and the following 4 days.

Subsequent to the second surgery, both experimental groups received once-daily subcutaneous administration of Teriparatid 0.4 µg/kg for 42 days.

Eight weeks after the second surgery, the animals were sacrificed.

### 2.6. µCT Scan Evaluation

Four different µCT scans were taken of each operated femur. A first µCT scan was performed immediately before the second surgery to confirm the creation of a non-union (pre-op scan). Two CT scans were performed at 4 and at 8 weeks after the second surgery to compare the evolution of the healing process (in vivo scans). The last scan was taken after extraction of the left femora from the sacrificed animals to evaluate the endpoint at a higher resolution (ex vivo scan).

All animals were scanned with a SkyScan 1076 in-vivo micro-computed tomography scanner (Brucker micro-CT, Kontich, Belgium) as previously described [24]. The test subjects were scanned with a scan orbit of 360 degrees, an isotropic pixel size of 18 μm, and energy settings of 100 kV (voltage), 280 ms (exposure time), and 100 μA (current) through a 1.0-(mm) aluminum filter. Although all the above settings remained the same for every µCT scan, the rotation step and the frame averaging were adapted regarding the different types of scans as follows (Table 1): rotation step 1°/frame averaging 2 (pre-op scan), rotation step 0.6°/frame averaging 4 (in vivo scans), rotation step 0.4°/frame averaging 6 (ex vivo scan).

SkyScan NRecon software (v.1.6.9.8, Brucker microCT, Konitch, Belgium) was used to perform image reconstruction. In the software, parameter settings of ring artefact reduction = 9/20, beam hardening correction = 30%, and smoothing = 1/10 were applied. The contrast limits were set according to the scan type, 0–0.035 for the pre-op and the ex vivo scan and 0–0.3 for the in vivo scans.

### 2.7. Quantitative Analysis

The SkyScan CTAnalyzer software (v.1.13.21, Brucker microCT, Kontich, Belgium) was used to perform a quantitative analysis of the datasets. The volume of interest (VOI) was generated for each scan individually. First, the longitudinal borders of the VOI were defined as 3 mm distal to and 3 mm proximal to the defect center as determined visually. This resulted in a VOI spanning 6 mm across the defect and consisting of 351 images. The transversal extent of the VOI was defined semi-manually with the built-in region of interest (ROI) function.

An ROI was manually drawn at intervals of ~15 slices. The ROI was drawn around the bone, leaving a narrow seam of soft tissue around the femur, ensuring that the entire volume of the bone was included. ROIs for the in-between images were calculated automatically by interpolation. The resulting ROIs were inspected individually, and new hand-drawn ROIs were added if needed. In the defect gap, where there was no bone tissue, ROIs were left empty. For the intervention group, a second set of ROIs was drawn, including the defect gap, by interpolating between the two ROIs from the bone ends, extending to the maximum width of the bone.

The VOIs generated for the control group will be further referred to as VOI_control. For the intervention group, three different VOIs were created for each scan.

A first VOI including bone, HACaS+G, and soft tissue inside the defect gap (VOI_all) was generated. Within the VOI_all a second VOI only consisting of bone was defined (VOI_bone, the equivalent to VOI_control). A third VOI subtracting the VOI_bone from the VOI_all was created (VOI_cer). The VOI_cer hence represented the mixture of HACaS+G and soft tissue in the defect gap. The analysis for the intervention group was run on each VOI separately.

Bone morphometry of the VOIs of the 4- and 8-week in vivo scans was performed, with thresholds set to 90 (lower) and 255 (upper) and using the despeckle function (despeckles ≤ 30 voxels). The parameters collected were bone volume, bone surface, bone surface density, trabecular number, and total porosity.

Two types of density protocols, bone mineral density (BMD) and tissue mineral density (TMD), were executed on the 4- and 8-week in vivo scans and ex vivo scans, respectively (Table 2). BMD was defined as the density of the whole VOI, and TMD as the density of the VOI after applying a threshold to exclude irrelevant soft tissue and liquid. The threshold for the TMD analysis was adapted to the quality of the scan, with 90–255 for the in vivo scans and 80–255 for the ex vivo scans. BMD was analyzed without a threshold except for the VOI_cer. For that, a 1–255 threshold was defined to exclude a black margin around the VOI artificially generated through the subtraction process before. In the intervention group, density analysis was performed only on the VOI_cer and VOI_bone, not on VOI_all.

### 2.8. Sacrifice

At 8 weeks after the second intervention, the animals were sacrificed under general anesthesia with CO_2_. The left femora were dissected under sterile conditions. The soft tissue was removed from the bones, and microbiological samples were taken. The animals were randomly assigned for histological or biomechanical testing. In the animals assigned to biomechanical evaluation, the right femora were dissected additionally.

After ex vivo µCT scanning, the bones assigned to biochemical testing were stored at −20 °C until further processing.

The bones intended for histological testing were directly fixated in 4.5% paraformaldehyde (Roti-Histofix, Roth, Karlsruhe, Germany) for 4 days.

### 2.9. Mechanical Testing

Two hours before biomechanical torsion testing, the femora (control group: 7 samples; intervention group: 7 samples) were left to thaw at room temperature in saline solution. The plate osteosynthesis was removed.

Biomechanical torsion testing was performed as previously described [24]. The proximal and distal ends were placed in two embedding molds (Technovit 4071, Heraeus Kulzer GmbH, Germany) using a fixture device. The lower embedding mold was connected to a pivotable axis while rotation of the upper mold was restrained. A linear, constant rotation (20°/min) was applied by the biomechanical testing device. The resulting maximum torque was recorded (8661-4500-V0200, Burster, Germany). Recording was stopped when the specimen fractured, if the maximum torque of 0.5 Nm was reached, or if the measured torque showed a stable decrease. The contralateral femora were tested identically.

### 2.10. Histology

After fixation of the specimens in 4.5% paraformaldehyde (Roti-Histofix, Roth, Karlsruhe, Germany) they were decalcified for ~3 weeks in ethylenediaminetetraacetic (Entkalker Soft, Roth, Karlsruhe, Germany) until the bone had softened sufficiently to allow slicing. The plates and screws were withdrawn, and the samples underwent a graded alcohol series for dehydration. Afterwards, the samples were embedded in paraffin. The femora were sliced at 5 µm thickness down to the center of the sample. Different histological staining methods were used, including iron hematoxylin (Carl Roth GmbH & Co. KG, Karlsruhe, Germany), brilliant-crocein-fuchsin acid, and safran (Pentachrom: Chroma-Waldeck GmbH & Co. KG, Münster, Germany), as well as tartrat-resistant acid phosphatase (TRAP) staining (Merck KGaA, Darmstadt, Germany). In addition, Immunohistochemical staining was performed using anti-CD14 (ab203294, Abcam, Cambridge, UK), anti-CD31 (ab182981, Abcam, Cambridge, UK), and anti CD68 (ab125212, Abcam, Cambridge, UK) antibodies. Whole slide images were generated. The slides stained with TRAP and Toluidine blue, and those stained immunohistochemically were analyzed quantitively via Fiji ImageJ as follows: A 6 mm ROI equivalent to the CT-VOI_control was selected, and the background was subtracted using a rolling-ball algorithm. A simple color threshold was employed to segment the image. For each image, two thresholds were set, one to segment the whole area of the bone and another to segment the areas specifically stained by each staining method. To obtain the most accurate color threshold possible, all histological slides were tested out separately and adjusted to ensure it was applicable for all slides of each staining method and to minimize oversegmentation. The total area of the resulting segments was measured using the built-in measure function of Fiji ImageJ.

All samples were provided by the Tissue Bank of the National Center for Tumor Diseases (NCT) Heidelberg, Germany, in accordance with the regulations of the tissue bank and the approval of the ethics committee of Heidelberg University.

### 2.11. Statistical Analysis

Data were recorded in Excel (Microsoft, Redmond, WA, USA) and statistically analyzed via Graphpad Prism version 9.1.0 (GraphPad Software, San Diego, CA, USA). The D’Agosino–Pearson test was employed to evaluate for normal distribution. To test for the statistical significance of differences between the four groups (Control; HACaS+G; PTH+HACaS+G; PTH) one-way analysis of variance (ANOVA) followed by Tukey’s multiple comparisons test were performed. The µCT data at the two time points (4 weeks and 8 weeks postop) were compared using Student’s paired *t*-test. *p*-values < 0.05 were considered statistically significant. All tests were performed two-sided. Data are presented as mean ± standard deviation (SD) in the figures and throughout the manuscript, unless otherwise indicated.

## 3. Results

Failure parameters:

In the PTH group, 13 out of 15 rats completed the study protocol. One rat died during anesthesia during the first intervention. A second rat had to be euthanized due to unconscionable wound conditions.

In the PTH+HACaS+G group, 13 out of 19 rats completed the study protocol. Four rats died during anesthesia during the second operation, one rat died during anesthesia during the 4-week post-op µCT scan and one animal had to be euthanized due to a severely reduced general condition after the intervention.

The animals that did not complete all steps of the protocol were excluded from all analyses.

In the PTH group, one ex vivo-µCT scan was excluded from analysis due to poor scan quality.

### 3.1. Neither PTH Nor PTH+HACaS+G Led to Stable Union of the Bone Defect

Biomechanical stability of the femora was tested by measuring the maximum torque they could withstand. The maximum torque values of the operated femora from both the PTH group and the PTH+HACaS+G group were significantly smaller than the contralateral intact femora (CF Control) (Figure 1). There was no difference between the PTH group and the PTH+HACaS+G group (Figure 1). Additionally, comparison of these groups to groups from a different set of experiments lacking PTH administration yielded similar maximum torque values. Thus, neither PTH administration alone nor the combination treatment of PTH and the bone graft substitute HACaS+G achieved functionally relevant stability of the bone, and no additional effect of PTH administration was observed.

### 3.2. µCT Analysis of the Bone in the PTH+HACaS+G-Group Showed an Osteoinductive Effect of the Bone Graft Substitute

In 3D analysis, an osteoinductive effect of the bone graft substitute could be seen. In the HACaS+G and PTH+HACaS+G-group percent bone volume and bone surface density were significant higher, whereas trabecular separation and total porosity were lower compared to the control group and the group solely treated with PTH. This could be observed both at 4 weeks and at 8 weeks (Figure 2a,b). The results of the 3D analysis did not differ in the groups treated with PTH compared to the groups without PTH.

To evaluate bone density, image segmentation was performed on the CT scans and a tissue density analysis was performed in the segmented regions corresponding to bone tissue following the tissue mineral density (TMD) protocol of the analysis software. The resulting bone mineral density (BMD) unit approximates g/cm^3^ for calcium hydroxyapatite. Bone density was significantly lower in the group treated only with PTH than in any other group. Between the other groups no significant differences were found (Figure 3).

### 3.3. µCT Analysis Revealed Biologic Remodeling of the Bone Graft Substitute HACaS+G in Presence of PTH

In the 3D analysis of the bone graft substitute, a biological structural remodeling of the bone graft substitute was imposed, which took place between the 4-week scans and the 8-week scans, as shown by a significant decrease in volume and surface volume ratio and an increase in total porosity (Figure 4b). This could be observed in both groups treated with HACaS+G. The relative decrease in bone graft volume was significantly lower in the group treated with PTH+HACaS+G. Other overt effects of PTH on the radiological properties of the bone graft substitute were not detected. Analogous to bone density analysis, density analysis of the bone graft substitute was performed. Bone graft mineral density was not significantly different between HACaS+G and PTH+HACaS+G at 8 weeks after implantation (Figure 4a).

### 3.4. PTH Enhanced the Osteoconductive Effect of HACaS+G

#### 3.4.1. Histological Staining Showed Noticeable Cell Invasion in PTH+HACaS+G

Pentachrome staining showed clear differences between the defect and the bone ends. HACaS+G could be identified easily as grey material between the bone ends. In both the native defect and the defect filled with HACaS+G, the immigration of different cells could be observed. Green, supposedly cartilage matrix, could especially be observed in both defect types (Figure 5).

TRAP staining used to evaluate the osteoclast population in the femora showed osteoclasts at the border between bone ends and defect. No significant differences between groups could be detected in quantitative analysis of the osteoclasts; there was, however, a tendency towards a larger number of osteoclasts in the PTH+HACaS+G group than in the group solely treated with HACaS+G (Figure 6).

Furthermore, the TRAP staining confirmed the persistence of the defect between the bone ends already observed in the pentachrome staining.

#### 3.4.2. CD14 and CD68 Staining Supported an Osteoconductive Effect of the HACaS+G

Both CD14 and CD68 are immunohistochemical markers for macrophages. The bone graft substitute in the PTH+HACaS+G group contained a similar amount of CD14^+^ and CD68^+^ cells as the HACaS+G group, indicating a comparable remodeling of the bone graft substitute. No additional effect of PTH administration upon invasion by these cells was found when comparing to the experimental group that received no PTH (Student’s paired *t*-test) (Figure 7).

#### 3.4.3. PTH Led to Increased Vascularization of the Defect and HACaS+G

The endothelial cell marker CD31 was used to quantify vascularization in the defect. Strikingly, when comparing the HACaS+G group treated with PTH and the PTH+HACaS+G group, the amount of CD31+ cells was significantly increased in the PTH group (Figure 8).

## 4. Discussion

The present study found that systemic administration of PTH supports the vascularization of the bone defect in a model of non-union. Additionally, implantation of HACaS+G resulted in an osteoinductive and osteoconductive effect.

We employed a two-step non-union animal model [25] to investigate a combinatory treatment of HACaS+G and PTH in addition to the standard surgical treatment of non-unions. Both treatment options, PTH and HACaS+G, have been tested extensively on several orthopedic indications, but to our knowledge, never in a combinatory treatment specifically on a two-step non-union model [11,12,13,14,19,20,21,26]. Our two-step model enabled us to test the combinatory treatment directly on non-union as opposed to acute fractures. This seems desirable for translating the tested methods into clinical non-union treatment.

Biomechanical stability is one of the requirements for a clinically successful non-union therapy [6]. It is important to note that we did not expect HACaS+G alone to achieve this. As PTH has been shown to have surprisingly strong effects on bone growth in animal models [19,20,21,26], we tested whether PTH alone or the combination of PTH and HACaS+G could achieve a stable union. Neither the combination of PTH and HACaS+G nor PTH alone resulted in a maximum torque value indicative of stability improvement compared to the other experimental groups. Nevertheless, our study focused on analyzing the healing process itself rather than confirming union at the endpoint.

The CT analysis of the bone ends adjacent to the defect yielded no positive effect of PTH administration on key radiological features such as bone density, contradicting the results from previous studies on PTH and fracture healing [26,27,28,29]. Combination treatment of the bone graft substitute HACaS+G and PTH, however, resulted in an osteoinductive effect, measured in an increased percent bone volume, bone surface density, and degree of anisotropy, as well as a decreased total porosity. This effect was most likely independent of PTH administration, as HACaS+G implantation alone resulted in similar values.

So far, animal studies have shown promising results after PTH treatment for fracture healing [18]. Particularly, an increase in density, bone volume, and stability was observed [30]. This was not reproduced in our study.

One explanation for this discrepancy between our study and past animal studies could be the catabolic effect of PTH. PTH is known to have both anabolic and catabolic effects [31]. A catabolic effect would be expected at continuously high PTH dosages, as seen in patients with hyperparathyroidism [32]. However, the PTH dosage in our study was lower than in some previous ones, which observed an overall anabolic effect—making it unlikely that we mimicked hyperparathyroidism.

It is possible that our PTH dosage was too low for a detectable effect on bone metabolism, especially in this non-union model, as it was considerably lower than in the mentioned studies (0.4 µg/kg/d vs. e.g., 60 µg/kg/d and 200 µg/kg/d) [20,21,27,28]. Although these experiments mostly resulted in clear effects of PTH on bone healing, the dosages were far larger than the dosages possible to administer to human patients. The standard daily dose for an adult human of 80 kg as prophylactic treatment in osteoporosis is 20 µg, respectively, 0.25 µg/kg/d. Higher dosages are not recommended in human subjects as they are associated with a higher incidence of adverse events [16]. PTH dosage in the present was chosen according to Leiblein et al. (2020), who found a positive effect of PTH administration on acute fracture healing after the administration of 0.4 µg/kg/d [18]. The benefit of the dosage we and Leiblein et al. [18] used is that it mirrored what could be realistically prescribed to a human patient [16,30]. This could also explain the incongruence between animal experiments and clinical studies: The clinical studies on PTH administration in fracture healing led to more ambiguous results than the experimental ones [30,33,34,35].

Additionally, size and complexity of the defect play a role in how much the healing process can be improved by PTH administration [36,37]. One study experimenting on critical size defects found an increase in bone density, but no increase in bone area and thus bone growth in rats treated with systemic PTH at 40 µg/kg/d [37]. Another study found that PTH was more effective at increasing bone density and volume in closed fractures than in an open fracture model [36]. Such studies on more complex fracture models are unfortunately the best approximators for non-unions, as data on PTH for non-union treatment is scarce and mostly gained from case reports [38,39]. Only one study tested adjuvant PTH treatment on non-unions in 32 patients, which demonstrated a good outcome, but lacked a control group [40]. The present study is to our knowledge the first to test PTH administration on a specific non-union animal model rather than an acute defect model. The possibility remains that the non-union situation itself is responsible for a PTH effect of a smaller degree.

Another important observation was that the radiological density of the implanted HACaS+G material (excluding bone and surrounding tissue) decreased throughout the course of the post-implantation observation period, prompting the hypothesis that during the healing period cells migrated into the bone graft substitute and started to remodel it, altering the radiological properties. A 3D μCT analysis further supported this finding of a potential osteoconductive effect. Interestingly, PTH administration reduced the decrease in bone volume throughout the experimental period. It seems possible that PTH promotes remodeling of the bone graft substitute to the bone, which would explain the smaller decrease of bone graft volume between 4 and 8 weeks in the PTH+HaACaS+G group, which, remarkably, was not accompanied by higher BMD (as would have been expected in the case of reduced resorption of the bone graft substitute). This theory is further supported by the histological analysis of the TRAP staining: Although no significant differences could be observed, a clear tendency toward increased TRAP staining in the PTH-HACaS+G-group indicates the activation of osteoclasts.

The quantitative analysis of the monocyte markers CD14 and CD68 revealed no significant differences between groups, which provides additional evidence for the physiological invasion of HACaS+G by cells like a native callus, as indicated by the µCT finding. This was further supported by the visual inspection of the pentachrome stained slices, where the cell migration into the HACaS+G and the beginning remodeling process of the bone graft substitute could be clearly observed. Furthermore, we observed no significant effect of PTH on monocyte number in the defect.

In TRAP staining, a tendency towards more osteoclast activity in the group treated with PTH and HACaS+G compared to either HACaS+G or PTH treatment alone could be observed, possibly hinting towards a synergistic effect of both treatments regarding osteoclast activation and thus bone remodeling.

The endothelial marker CD31 was present in similar quantities in bone graft substitute and native callus, which demonstrates equivalent vascularization of the bone graft substitute compared to native callus.

Remarkably, PTH administration caused a dramatic increase in CD31^+^ cells and thus supported vascularization in the defect. This was the case in both PTH mono treatment as well as in combination with HACaS+G.

Vascularization is an essential part of the bone healing process [41,42]. So far, several studies have shown an impact of PTH on vascular growth in bone.

Langer et al. [43] found that PTH administration of 100 µg/kg/d clearly influenced the vascular growth in bones in rats [43]. An experiment on mice showed an increased density of CD31+ blood vessels after short time intermittent PTH administration of 43 µg/kg/d [44]. In a mouse-fracture model, daily 40 µg/kg/d PTH administration over 14 days increased vessel number and CD31^+^ expression in fracture callus [45].

Furthermore, in the PTH+HACaS+G-group, CD31^+^ cells were observed in the implanted material in a quantity similar to the number of CD31^+^ cells in the native callus in the PTH group, which had no HACaS+G implanted. Thus, PTH administration can achieve the vascularization of implanted bone graft substitute.

## 5. Conclusions

We observed an osteoinductive and osteoconductive effect of HACaS+G on the treatment of non-unions. Unlike the previous reports on general PTH action and PTH in acute fractures, PTH does not appear to have an osteoinductive effect in non-unions. Nevertheless, our finding is in line with previous reports insofar as a positive effect of PTH on vascularization was observed. Therefore, PTH could be an attractive option for a combination treatment after bone graft substitute implantation. Additionally, the PTH dosage used in the present study more closely resembles what is used in the clinical setting, suggesting that the mechanism of increased vascularization through PTH administration could be reproducible in clinical non-union therapy.

## Figures and Tables

**Figure 1 cells-10-02058-f001:**
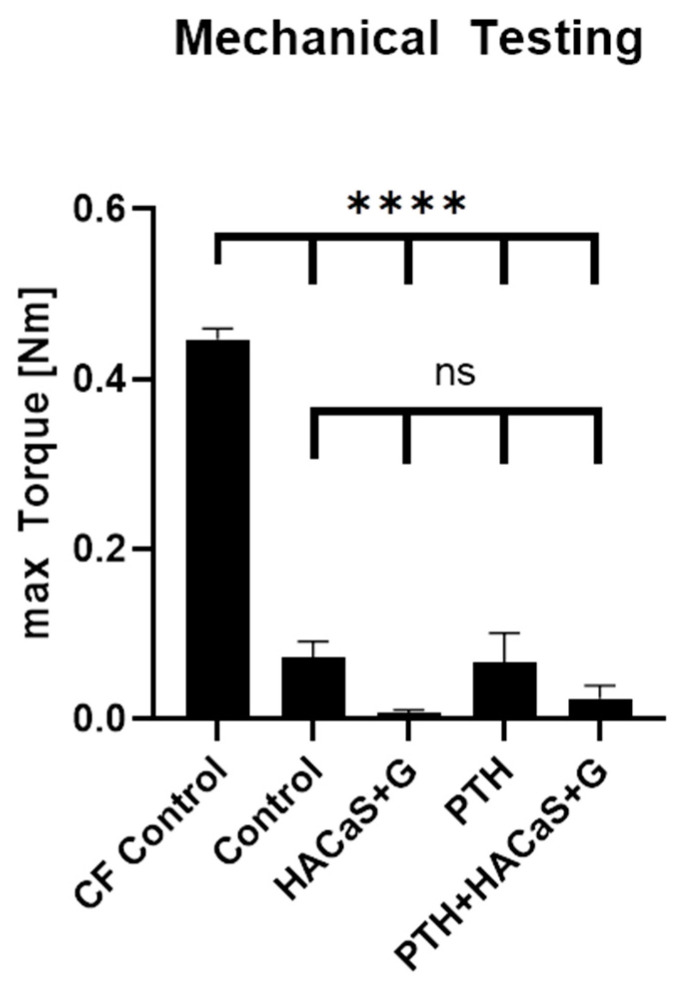
Mechanical testing. The max. torque of the contralateral intact femora (CF control) is significantly higher than all other groups, but there are no significant differences between the control and the treatment groups (PTH, HACaS+G and PTH+HACaS+G). **** = *p* < 0.0001, ns = not significant.

**Figure 2 cells-10-02058-f002:**
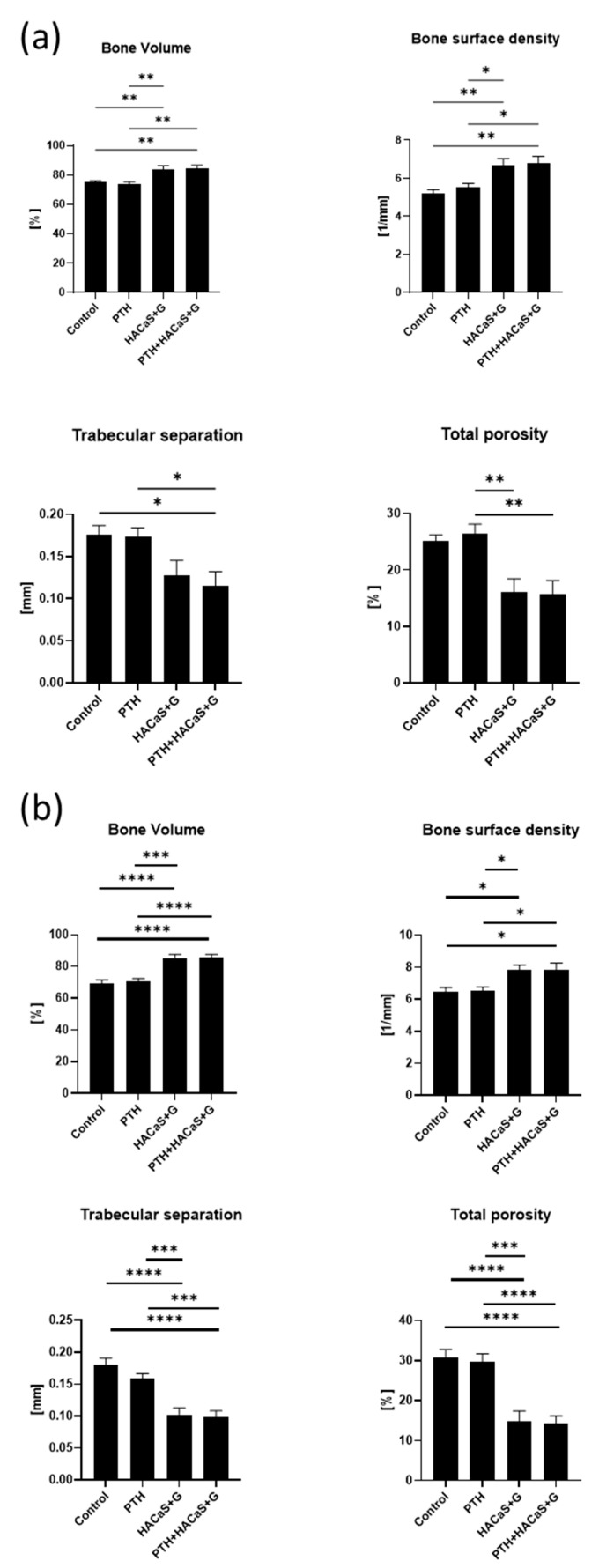
3D Analysis of bone tissue: (**a**) 4 weeks after the second surgery and (**b**) 8 weeks after the second surgery. * = *p* < 0.05, ** = *p* < 0.01, *** = *p* < 0.001, **** = *p* < 0.0001, ns = not significant.

**Figure 3 cells-10-02058-f003:**
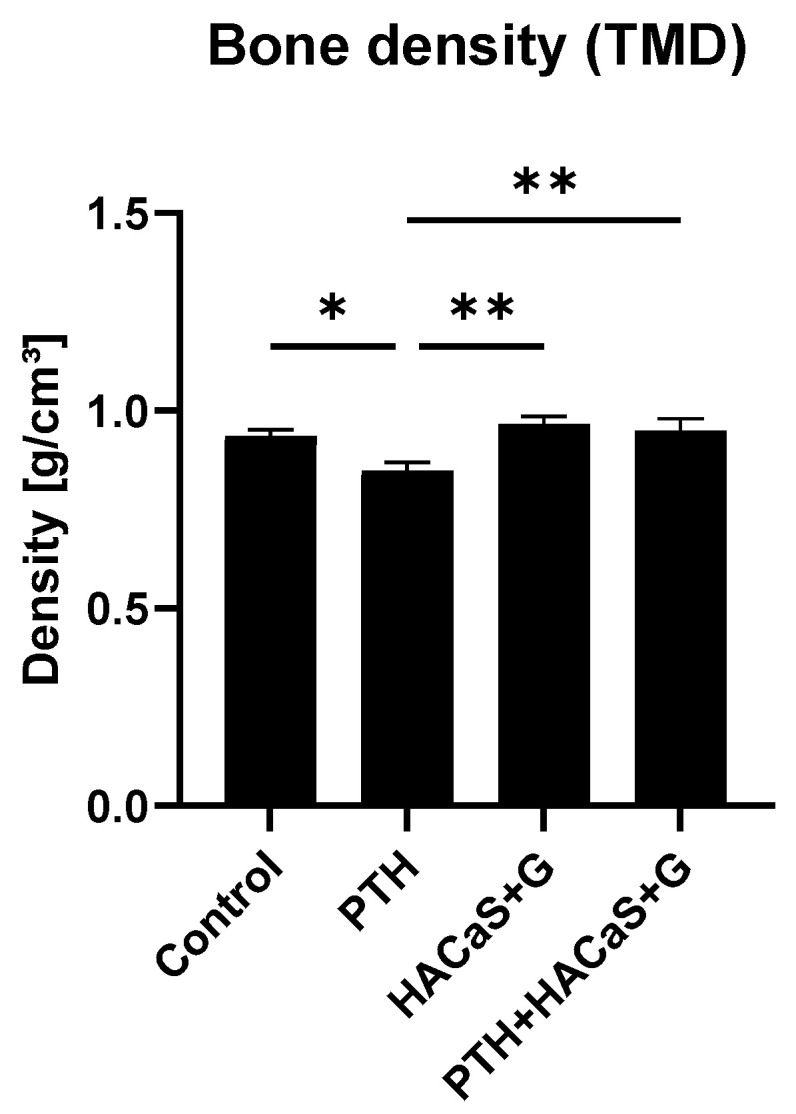
Analysis of bone density (TMD). PTH-treated samples showed less density in comparison to the other groups. * = *p* < 0.05, ** = *p* < 0.01.

**Figure 4 cells-10-02058-f004:**
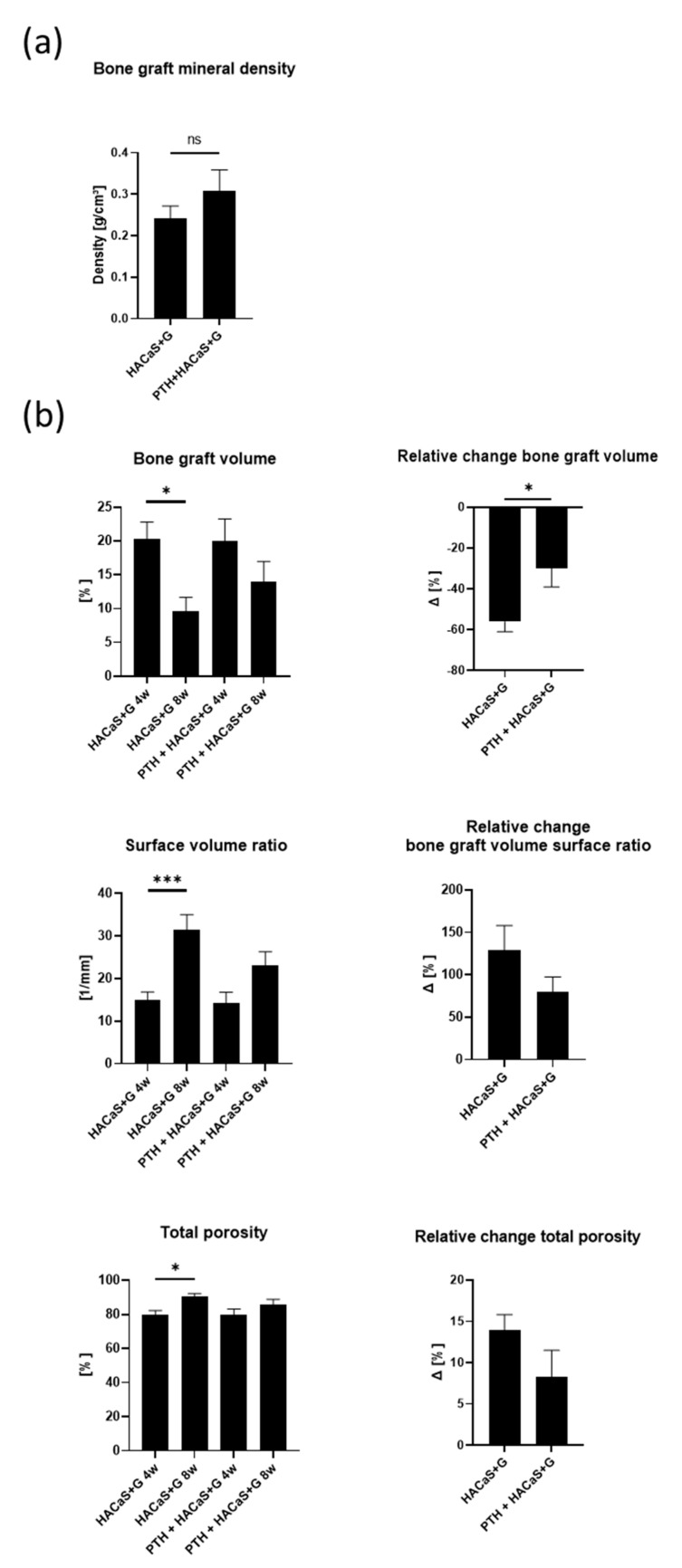
Analysis of bone graft substitute via µCT: (**a**) comparison of bone graft mineral density 8 weeks after insertion between HACaS+G and PTH+HACaS+G, and (**b**) comparison of 3D analysis of bone graft substitute 4 and 8 weeks after insertion. * = *p* < 0.05, *** = *p* < 0.001, ns = not significant.

**Figure 5 cells-10-02058-f005:**
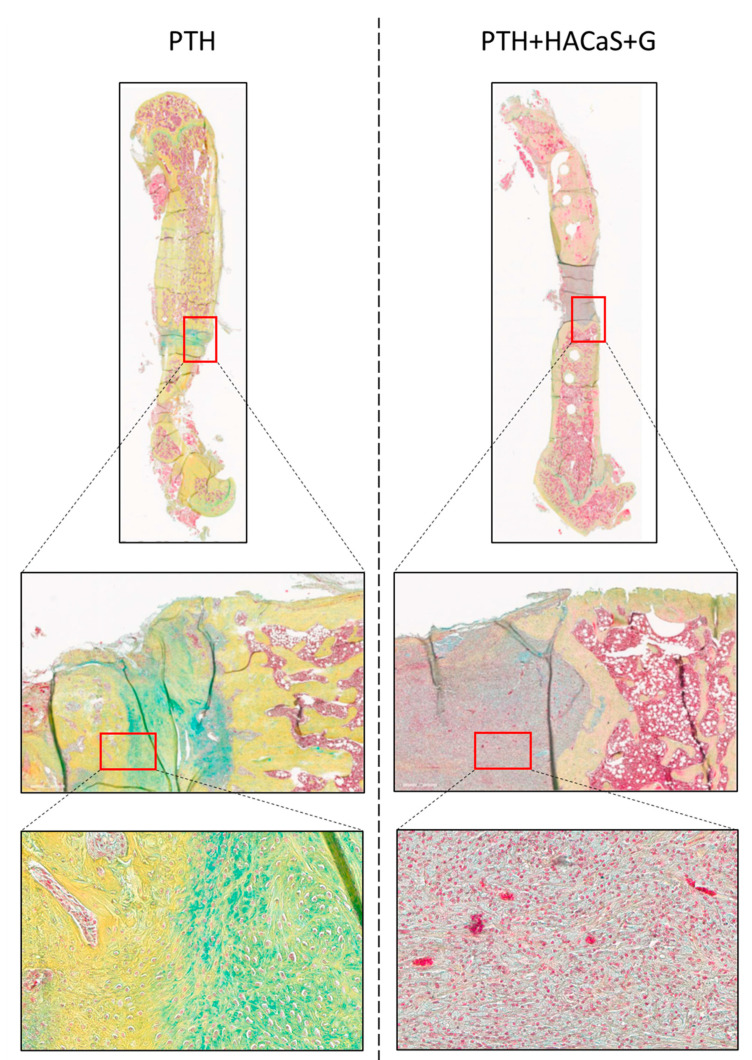
Pentachrome staining. Reddish cytosol in PTH+HACaS+G shows noticeable cell invasion in HACaS+G. Collagen and reticular fibers are stained yellow whereas green indicates cartilage matrix.

**Figure 6 cells-10-02058-f006:**
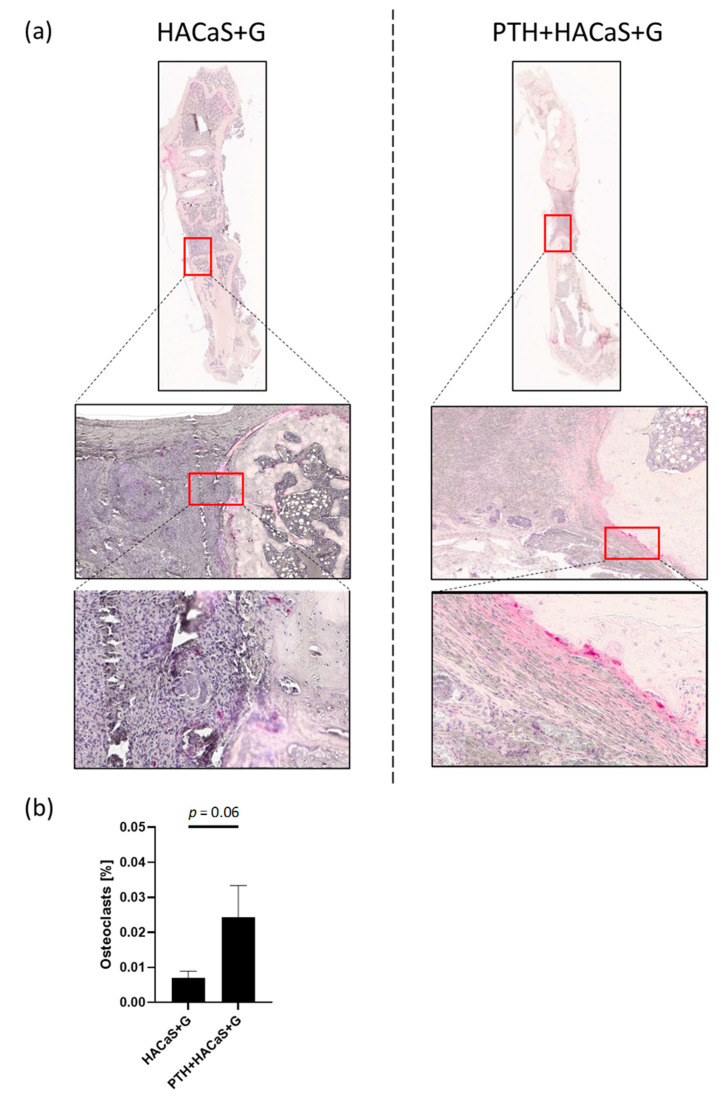
TRAP staining shows invasion of osteoclasts in HACaS+G: (**a**) TRAP staining (osteoclasts with red staining), and (**b**) quantification.

**Figure 7 cells-10-02058-f007:**
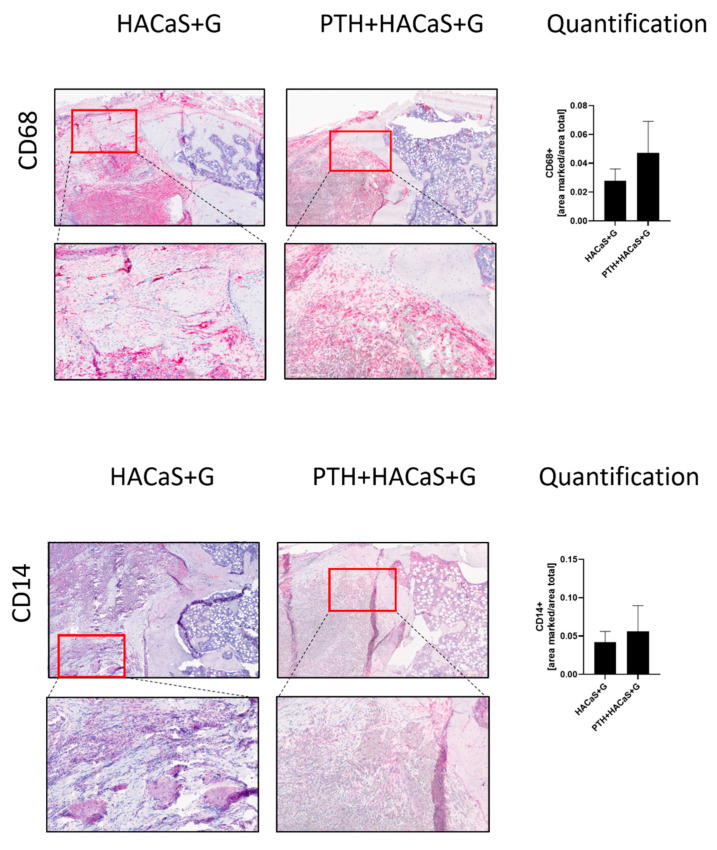
CD14 and CD68 staining, markers for tissue macrophages. This is a comparison of the outer edge of the defect gap between the HACaS+G- and the PTH+HACaS+G-treated groups. Some of the remaining boney ends can be observed towards the right side of the image.

**Figure 8 cells-10-02058-f008:**
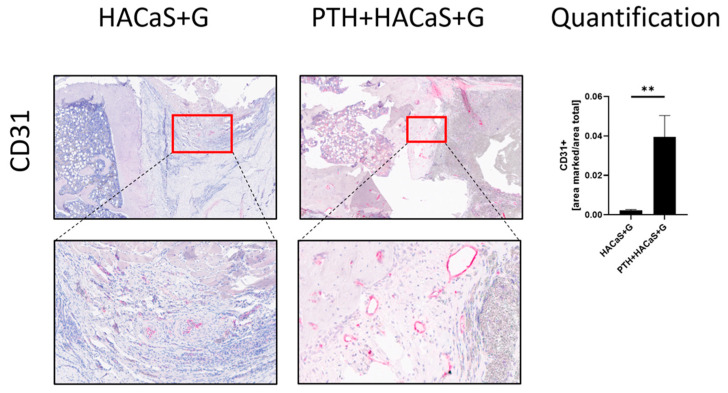
CD31 staining, a marker for vascularization. This is a comparison of representative parts of the defect gap between the HACaS+G- and PTH+HACaS+G-treated group. PTH leads to increased CD31 positivity in HACaS+G. ** = *p* < 0.01.

**Table 1 cells-10-02058-t001:** Scan settings.

Scan	Rotation Step	Frame Averaging
Pre-op scan	1°	2
4- and 8-week in vivo scan	0.6°	4
Ex vivo scan	0.4°	6

**Table 2 cells-10-02058-t002:** Thresholds for density evaluation.

		BMD	TMD
**In Vivo**	VOI_control	No threshold	90–255
	VOI_bone	No threshold	90–255
	VOI_cer	1–255	90–255
**Ex Vivo**	VOI_control	No threshold	80–255
	VOI_bone	No threshold	80–255
	VOI_cer	1–255	80–255

## Data Availability

The datasets used and analyzed during the current study are available from the corresponding author on reasonable request.

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
