# Peer review of "Systemic Administration of PTH Supports Vascularization in Segmental Bone Defects Filled with Ceramic-Based Bone Graft Substitute"

_cells, 2021, doi:10.3390/cells10082058_

Round 1

Reviewer 1 Report

The reported study compared the efficacy of a bone graft substitute (Cerament G) with and without intermittent PTH administration on outcomes of non-union bone defects repair in rats. The impact of treatments was assessed using micro-CT, histological analysis, and torsion testing. At the dose used, PTH had no beneficial effect on repair, but an impact of PTH treatment on vascularization within the defect was observed. The authors conclude that PTH administration could have beneficial effects as a co-treatment with bone graft substitutes.

The manuscript is well-written and the discussion is comprehensive.

Specific comments:

  1. The dose of PTH used is extremely low. The authors are aware of this fact which they raise in the discussion. At the dose used, it is interesting that they were able to observe changes in vascularization but the authors should exercise caution in concluding that the PTH treatment has no effect on bone healing; this could only be due to the low dose used.

  1. What is the rationale for using only female animals? This is not justified in the text.

  1. It is very difficult to interpret the images presented in Figures 5 and 6. Higher quality images should be provided with additional labeling of histoimmunochemistry signal and salient tissue structures.

  1. How is the expression of macrophage markers an indication of an osteoinductive effect? The infiltration of macrophages happens at the beginning of the repair process. What is the relevance of the CD14 and CD68 staining at the end of the experiment? Labeling for osteoblast markers would seem more appropriate.

Minor comment:

There are a few formatting and typographical errors:

-p.2, line 79: the strikethrough text should be deleted;

-p.2, line 121: Forteo, not Forsteo;

-p.2, line 121: Teriparatide, not Teriparatid;

-p. 10, Figure 2: the A and B panels are not labeled;

-p. 16, lines 466 and 468: hyperparathyroidism, not hyperparathyreodism.

Reviewer 2 Report

The manuscript of Freischmidt et al. “Systemic administration of PTH supports vascularization in 2 segmental bone defects filled with ceramic-based bone graft 3 substitute” is a nice study raising an important issue of the influence of PTH and bone graft substitute inhibitor on bone healing. The authors describe recently established sequential defect model, which provides a platform to test new potential therapeutic strategies on non-unions. According to the results described in the paper the effects of a combinatorial treatment of a bone graft substitute implantation and systemic PTH administration was assessed by a variety of methods including μ-CT, histological analysis, and bio-mechanical testing and compared to monotreatment and controls. Thie authors suggest that neither PTH alone nor the combination of a bone graft substitute and PTH led to the formation of a stable union, but the data demonstrate a clear osteoinductive and osteoconductive effect of the bone graft substitute.

The manuscript is very well and clear written, the methods are well justified. There are minor concerns before considering the publication of this study.

  1. The choice of the particular journal is quite inappropriate as real cell biology does not appear in the manuscript. May be, it would be wise to transfer the manuscript to some more suitable journal of the family?
  2. Figure 1 appears damaged in the pdf version available to the reviewer and should be corrected
  3. Figure 7 and Figure 8 show only pathological tissue. A picture of a normal tissue would be supporting and clarifying.
  4. The first passage in the discussion section absolutely repeats the corresponding section of the Introducton and should me omitted.
  5. The whole study would benefit if the authors slightly shorten both Introduction and Discussion and emphasise the presise novelty and compare with similar works.

Reviewer 3 Report

Dear Editor and Authors,

the manuscript “Systemic administration of PTH supports vascularization in segmental bone defects filled with ceramic-based bone graft substitute “ by Freischmidt et al. evaluates the effect of a combination treatment of non union fracture healing process.  

The authors employed a previously developed animal model tailored for the study of infection-related non-unions with segmental bone defect (Helbig L, et. al BMC Musculoskelet Disorders. 2020) to evaluate the effect of PTH administration. In particular, the authors hypothesized that PTH combined with bone substitute would have had positive effect on bone healing and vascularization in the treatment of non-union. The effects of the treatment was monitored looking at micro-CT, biomechanical testing and  histochemical analysis. The two stage animal non union model revealed that neither PTH nor PTH+HACa+G led to stable non union of bone defect. However, PTH enhanced vascularization of the bone defect.  

The animal model developed by the authors offered a biological platform for the screening molecules which positively support bone graft substitutes for new therapeutical approaches. The development of new therapy of delayed or non-union healing is an area of scientific soundness thus it is of interest to the readers. This model could be valuable to gather reliable data on the underlying mechanism of fracture healing to the treatment method. However, the authors reported an approximate description of the cell-types which migrate to the site of healing without deepen the cell function.

Overall, the paper is well written and provides sufficient background. The results are clearly presented however the research design could be improved. The referee believes that author choice of testing PTH was meaningful, nonetheless the referee does not see why such a low administration dose was employed. Please justify.

 Minor points:

pg7 line282. Please specify how the quantification has been performed

pg10 : insert the letters a and b within the figure 2  

pg13: line 382.  the referee does not agree that the cell invasion in PTH+HACaS+G is massive

pg16 : line 450. As mentions above, to the referee the authors do not report any information about cell functions. Please remove functional indicators.

Round 2

Reviewer 1 Report

The authors have appropriately addressed the reviewers' comments. The manuscript is improved.